# Efficient Fourier Single-Pixel Imaging with Gaussian Random Sampling

**Ziheng Qiu [1], Xinyi Guo [1], Tian'ao Lu [1], Pan Qi [2], Zibang Zhang [1,3,]\* and Jingang Zhong [1,3]**

[1] Department of Optoelectronic Engineering, Jinan University, Guangzhou 510632, China; qiuziheng@stu2019.jnu.edu.cn (Z.Q.); sydneeguo@stu2016.jnu.edu.cn (X.G.); lutianao1994@stu2017.jnu.edu.cn (T.L.); tzjg@jnu.edu.cn (J.Z.)

[2] Department of Electronics Engineering, Guangdong Communication Polytechnic, Guangzhou 510650, China; qiqipan@gdcp.edu.cn

[3] Guangdong Provincial Key Laboratory of Optical Fiber Sensing and Communications, Jinan University, Guangzhou 510632, China

\* Correspondence: tzzb@jnu.edu.cn

**Abstract:** Fourier single-pixel imaging (FSI) is a branch of single-pixel imaging techniques. It allows any image to be reconstructed by acquiring its Fourier spectrum by using a single-pixel detector. FSI uses Fourier basis patterns for structured illumination or structured detection to acquire the Fourier spectrum of image. However, the spatial resolution of the reconstructed image mainly depends on the number of Fourier coefficients sampled. The reconstruction of a high-resolution image typically requires a number of Fourier coefficients to be sampled. Consequently, a large number of single-pixel measurements lead to a long data acquisition time, resulting in imaging of a dynamic scene challenging. Here we propose a new sampling strategy for FSI. It allows FSI to reconstruct a clear and sharp image with a reduced number of measurements. The key to the proposed sampling strategy is to perform a density-varying sampling in the Fourier space and, more importantly, the density with respect to the importance of Fourier coefficients is subject to a one-dimensional Gaussian function. The final image is reconstructed from the undersampled Fourier spectrum through compressive sensing. We experimentally demonstrate the proposed method is able to reconstruct a sharp and clear image of $256 \times 256$ pixels with a sampling ratio of 10%. The proposed method enables fast single-pixel imaging and provides a new approach for efficient spatial information acquisition.

**Keywords:** computational imaging; single-pixel imaging; sampling strategy; compressive sensing; optimization

## 1. Introduction

Single-pixel imaging [1–8] is a computational imaging technique that allows images to be acquired by using a spatially unresolvable detector, namely, single-pixel detector (such as, photodiode, solar cell, and photomultiplier tube). Compared with typical pixelated detectors (such as, CCD and CMOS), single-pixel detectors can work at a wide waveband, especially at the wavebands where pixelated detectors are expensive or even technically unavailable (such as infrared, deep ultraviolet, X-ray, or terahertz). Thus, single-pixel imaging has been considered as a potential solution for imaging at special wavebands and attracted a lot of attention in the last decade [9–16].

The key to single-pixel imaging is spatial light modulation. Spatial light modulation allows the spatial information of the target object to be encoded into a 1-D light signal sequence, which is suitable for single-pixel detection. The desired object image can be retrieved by decoding the spatial information from the resulting single-pixel measurements through the image reconstruction algorithm corresponding to the spatial light modulation strategy.

Fourier single-pixel imaging (FSI) [17–22] is a branch of basis-scan single-pixel imaging techniques. It uses Fourier basis patterns for spatial light modulation. Fourier basis patterns are also known as sinusoidal intensity patterns. The Fourier spectrum of the desired object image can be acquired by using Fourier basis patterns for structured illumination or structured detection. Compared with other basis-scan single-pixel imaging methods (such as, Hadamard [23–27], DCT [28]), FSI has been proven more data-efficient when the differential method of measurement is employed [29]. Specifically, FSI with 3-step phase shifting can reconstruct a lossless image in a differential-measurement manner taking as many measurements as 1.5 times the number of image pixels. For other basis-scan single-pixel imaging methods, differential measurement will result in the single-pixel measurements being doubled. Moreover, the generation of Fourier basis patterns is flexible. The basis patterns of FSI can be generated by the interference of two plane waves [30], which potentially allows FSI to be implemented without using a pixelated spatial light modulator. Such a property benefits imaging at the wavebands where spatial light modulators are not available.

However, as other single-pixel imaging methods do, FSI suffers from the tradeoff between imaging quality and imaging time. The spatial resolution of the image reconstructed by FSI mainly depends on the number of Fourier coefficients sampled. Specifically, it requires more spatial information to reconstruct an image with finer details. The more spatial information implies more single-pixel measurements, and consequently, longer data acquisition time. However, the data acquisition time is crucial for fast imaging, especially when imaging a dynamic scene. Thus, it is worth exploring how to improve the data efficiency in FSI.

Initially, FSI was proposed with the spiral sampling strategy [17]. The sampling strategy exploits the prior knowledge that most information of natural images is concentrated in low-frequency bands of the Fourier space. According to the spiral sampling strategy, only low-frequency Fourier coefficients will be sampled with high-frequency coefficients discarded. However, the lack of high-frequency components could result in severe ringing artifacts in the reconstructed images, especially when the sampling ratio is low. Later, several sampling strategies were reported, such as statistical-importance [18], diamond [31], circular [31], and polynomial [32]. Different sampling strategies are referred to different sub-sets and different orderings of the Fourier basis patterns used for spatial light modulation. We note that the research on basis patterns ordering is a hot spot in single-pixel imaging, because an optimal sampling strategy enables images of unchanged quality to be reconstructed from the least single-pixel measurements and therefore shortest data acquisition time. For example, Russian doll [23], cake-cutting [24], origami [25], and total variation ascending orderings [26] were recently proposed for Hadamard single-pixel imaging.

Here we propose a sampling strategy for FSI termed Gaussian random sampling. The core of the proposed sampling strategy is to perform a variable density sampling in the Fourier space and the density is based on the importance of Fourier coefficients. Specifically, the sampling density with respect to the importance of Fourier coefficients is subject to a 1-D Gaussian function. The importance of a Fourier coefficient is referred to the magnitude of the modulus of the coefficient. In other words, the larger the modulus of a Fourier coefficient is, the more important this coefficient is. Combined with compressive sensing (CS), the proposed method is able to reconstruct a clear and sharp image from far fewer single-pixel measurements than image pixels. We experimentally demonstrate the proposed method is able to reconstruct a high-quality image of $256 \times 256$ pixels with a sampling ratio of 10%. The proposed method enables fast single-pixel imaging and provides a new approach for efficient spatial information acquisition.

## 2. Principle

A schematic diagram of structured illumination-based FSI set-up is shown in Figure 1a. As the figure shows, a digital micro-mirror device (DMD) is used as the spatial light modulator to generate Fourier basis patterns. Each Fourier basis pattern can be expressed as

$$P(x,y) = \frac{1}{2} + \frac{1}{2} \cdot \cos\left[2\pi(f_x x + f_y y) + \varphi\right],\tag{1}$$

where $(x,y)$ denotes the coordinate in the spatial domain, $\varphi$ denotes the initial phase, and $f_x$ and $f_y$ are spatial frequency corresponding to $x$ and $y$ direction, respectively. As DMDs are capable of high-speed binary patterns generation, Fourier basis patterns are generally binarized through dithering [33], as the inset in Figure 1a shows. A photodiode amplifier (PDA) is used as the single-pixel detector to collect the back-scattered light from the object under structured illumination. The Fourier spectrum of the desired object image can be acquired by using the three-step phase-shifting strategy, as Figure 1b shows. Each Fourier coefficient, $\tilde{I}(f_x, f_y)$, is acquired by using a set of three Fourier basis patterns of the same spatial frequency pair but a different initial phase. The initial phase, $\varphi_i$, of the $i$-th step pattern is $2(i-1)\pi/3$. The Fourier coefficient $\tilde{I}$ associated with the spatial frequency $(f_x, f_y)$ can be calculated through

$$\tilde{I}(f_x, f_y) = (2D_1 - D_2 - D_3) + \sqrt{3}\mathrm{j}(D_2 - D_3),\tag{2}$$

where j is the imaginary unit, and $D_i$ denotes the single-pixel measurement corresponding to the $i$-th step pattern. As Figure 1b shows, the Fourier spectrum of a real-valued image is conjugate symmetric. Thus, the symmetric coefficients need not be sampled. To reconstruct a lossless image by FSI, the number of Fourier coefficients acquired is one half of the number of image pixels. If the three-step phase shifting strategy is employed for differential measurement, the number of single-pixel measurements will be 1.5-fold the number of image pixels. The object image can be reconstructed from the Fourier spectrum acquired through a 2-D inversed Fourier transform or CS. The proposed method uses CS for image reconstruction.

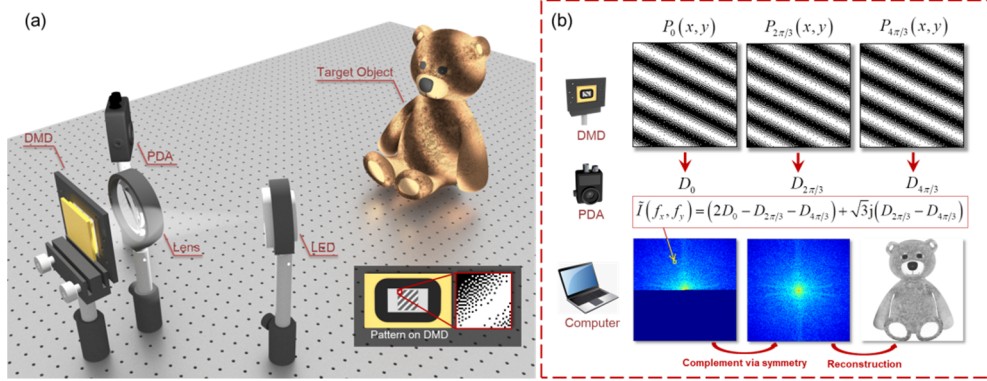

**Figure 1.** Illustration of three-step phase shifting FSI. (**a**) In a structured illumination-based setup, the object is under illumination of Fourier basis patterns generated by a DMD. The Fourier basis patterns are dithered. (**b**) The object image is retrieved by acquiring the Fourier spectrum of the image. Each complex-valued Fourier coefficient can be acquired by using a set of three-step phase-shifting Fourier basis patterns where the phase shift is $2\pi/3$. The conjugate symmetry of the Fourier spectrum allows a lossless image to be retrieved with only one half of the Fourier coefficients acquired.

The reconstruction of a large-size image requires a large number of single-pixel measurements, resulting in a long data acquisition time. Undersampling is a typically used strategy to reconstruct an image of satisfactory quality with a reduced number of measurements. In the context of FSI, undersampling means only a portion of the Fourier spectrum

is sampled. Inspired by the work by W. Meng et al. [32], we propose a Gaussian random sampling strategy for FSI. The key to the proposed sampling strategy is to perform a variable density sampling in the Fourier space and, more importantly, the density with respect to the importance of Fourier coefficients is subject to a 1-D Gaussian function. In other words, the more important the Fourier coefficient is, the higher probability the coefficient is to be sampled with. The final image is reconstructed from the under-sampled Fourier spectrum through CS. CS is referred to algorithms that can recover certain sparse or compressible signals or images (the length of the signals is $M \times N$) from far fewer samples or measurements (the length of the measurements is $n$ and $n << M \times N$) than traditional methods (according with Shannon's theorem) use. How to retrieve the signals from a far small number of sampled data is an ill-posed and under-determined problem. However, CS algorithms can recover sparse solutions by imposing a series of convex-optimization constraints, such as $l_p$–norm minimization, greedy algorithm, minimum total variation, etc.

However, it is difficult to predict which Fourier coefficients are important for any object or scene to be imaged. Here we adopt a statistics method reported by Bian et al. [18] to derive the importance distribution of coefficients in the Fourier space for reference. Specifically, we use DIV2K database [34], which provides hundreds of high-quality natural images. As Figure 2a shows, we use all 800 natural images from the database and each high-resolution full-color image first is converted into grayscale and segmented to a number of $M \times N$-pixel sub images, where $M \times N$ is the size of the reconstructed image. Then we apply a 2-D Fourier transform to every single sub image and sum up the moduli at the corresponding locations of all resulting Fourier spectra. Lastly, the Fourier coefficients of the summed Fourier spectrum in a size of $M \times N$ pixels are sorted in a descending order of magnitude. Please note that the conjugate symmetric coefficients are discarded. In our case, $M = 256$ and $N = 256$. The number of sorted coefficients is 32,770.

Each sorted coefficient has its own index, $k$, which indicates the importance of the coefficient. The smaller the index is, the higher the importance of the coefficient. Next, as Figure 2b shows, we generate a uniformly distributed random function $r(k)$ whose range is from 0 to 1. We also generate a Gaussian function

$$g(k) = \exp\left\{ -[(k-1)/k_{\max}]^2/\sigma \right\}, \tag{3}$$

where $k$ is a positive integer denoting the index with descending order of importance, $k_{\max} = 32{,}770$ when $M = 256$ and $N = 256$, and $\sigma$ is the standard deviation of the Gaussian function. The value of $\sigma$ depends on the sampling ratio $\eta$. Here, the sampling ratio is defined as twice the number of sampled Fourier coefficients to the number of total Fourier coefficient in the Fourier spectrum, where "twice" is for the conjugate symmetry. When $\eta < 0.5$, there is a simple relation between the standard deviation and the sampling ratio, that is, $\sigma = (2\eta)^2/\pi$. As indicated by the red lines in Figure 2b, the maxima of the 1-D Gaussian function is at $k = 1$. If $g(k) > r(k)$, then the $k$-th Fourier coefficient is marked to be sampled, and vice versa. The resulting sampling masks for different sampling ratios are shown in Figure 2c. Please note that white pixels in the masks indicate the Fourier coefficients at the corresponding locations are to be sampled.

We note that such a sampling strategy would result in a few high-importance coefficients not being sampled, but adopting a CS algorithm for image reconstruction allows those un-sampled high-importance coefficients to be recovered through optimization. It is because high-importance coefficients are sampled with a high density, which imposes a strong constraint to find the globally optimized solution for the un-sampled high-importance coefficients. As such, more single-pixel measurements can be spent in sampling the remaining low-importance coefficients and those low-importance coefficients mainly contribute to high-frequency information. Consequently, the spatial resolution of the resulting image is improved.

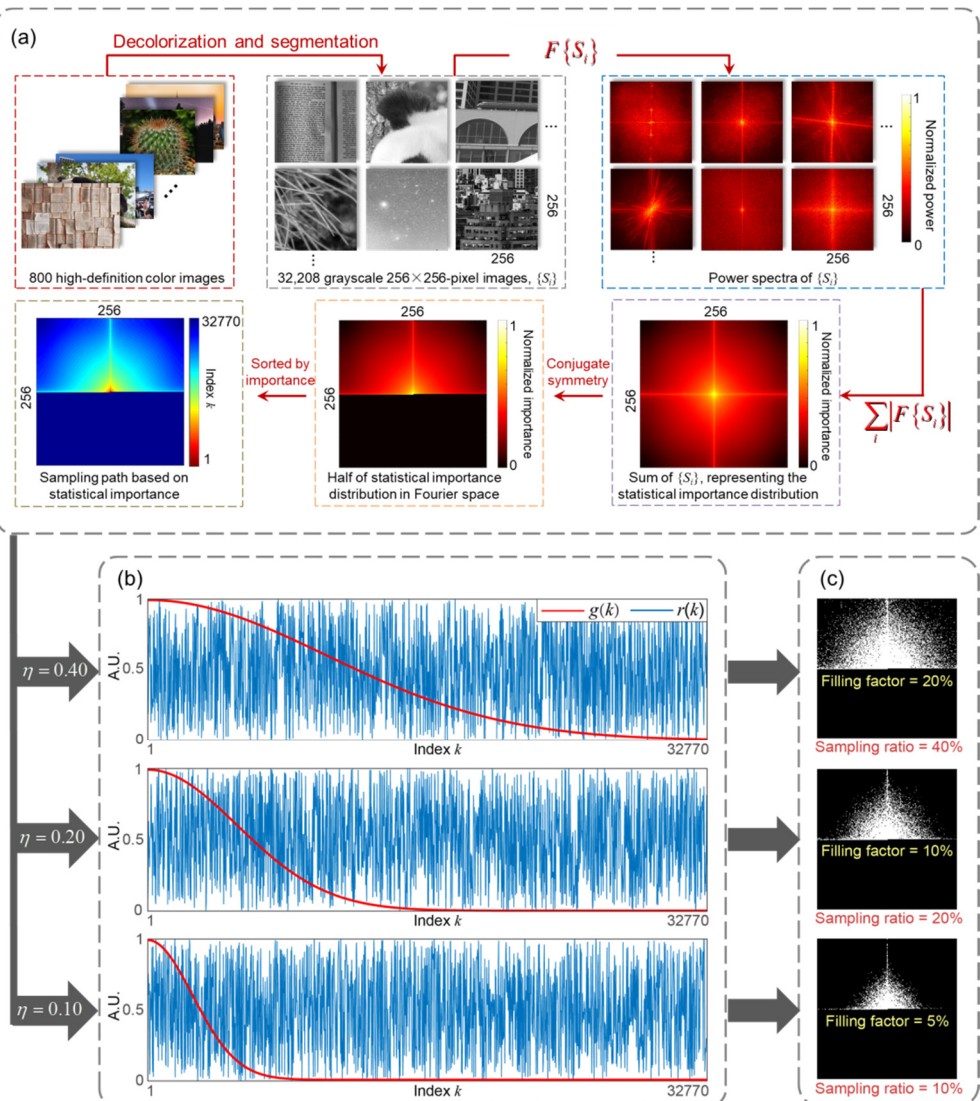

**Figure 2.** The generation of Gaussian random sampling mask by the proposed method. In the first step (**a**), the important distribution of coefficients in the Fourier domain is obtained through statistics. Images from a database are decolorized and segmented to $M \times N$ pixels. Fourier transform is applied to each segmented sub image and the moduli at the corresponding locations of the resulting Fourier spectra are summed up. Discarding the conjugate symmetric coefficients in the summed spectrum, the remaining coefficients are sorted in a descending order. Each sorted coefficient has its own index $k$, which indicates the importance of the coefficient. The smaller the index is, the higher the importance of the coefficient. In the second step (**b**), a uniformly distributed random function $r(k)$ and a Gaussian function $g(k)$ with a specific sampling ratio, $\eta$, are generated. In the third step (**c**), all Fourier coefficients, whose index $k$ satisfies $g(k) > r(k)$, are marked as 'to be sampled' (white pixel) in the sampling mask. Filling factor is defined as the ratio of marked coefficients to all coefficients in the Fourier space, which is also one half of the sampling ratio (due to the conjugate symmetry).

## 3. Simulation

The proposed method is first validated by numerical simulations. The simulations are conducted on a desktop computer equipped with an Intel(R) Core(TM) i7-8700K CPU, 16 GB RAM, Windows 10 operating system, and MATLAB 2019a. The CS algorithm we employ for image reconstruction is L1-Magic [35].

To demonstrate the advantages of the proposed sampling strategy, we compare it with another three methods, including radial sampling strategy [36], circular sampling strategy [31], and polynomial sampling [32]. The methods in comparison are either typ-

ically used or recently proposed. Figure 3 shows the sampling masks generated by the aforementioned strategies for different sampling ratios. In particular, the polynomial sampling strategy requires two user-defined parameters, $\rho$ and $R$. The combination of the two parameters, $(\rho, R)$ is set to be $(18, 0.05)$, $(9, 0.05)$, $(7, 0.1)$, and $(5, 0.18)$ for sampling ratios 1%, 3%, 5%, and 10%, respectively. These parameters settings guarantee the best performance of the polynomial sampling strategy in our case.

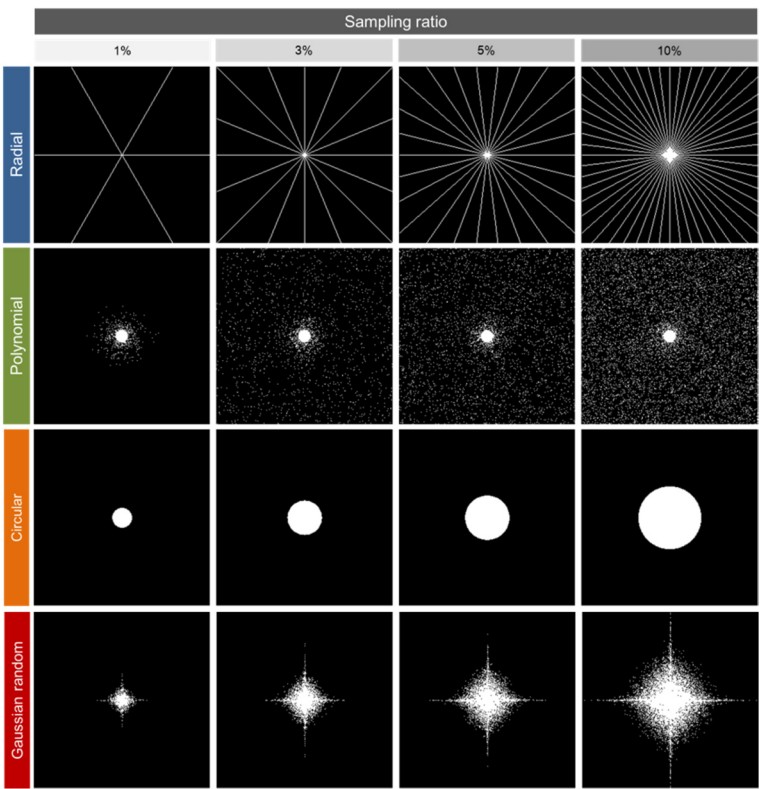

**Figure 3.** Sampling masks used in the simulations and the experiments.

In the first simulation, a USAF-1951 resolution chart pattern is used as the test image. The image is with $256 \times 256$ pixels. As the results show in Figure 4, the radial sampling strategy is not able to reconstruct any bars, when the sampling ratio is below 10%. Even when the sampling ratio is 10%, the finest resolvable bars are Group-2 Element 5. In addition, the circular sampling strategy can successfully reconstruct Group-2 Element 6, when the sampling ratio is 3%. The polynomial sampling strategy and the proposed sampling strategy can even reconstruct Group-1 Element 1 when the sampling ratio is 3%, but the image reconstructed by the polynomial sampling strategy appears blurred and smeared. When the sampling ratio is 5%, the circular sampling strategy can only reconstruct Group-1 Element 2, while the polynomial sampling strategy and the proposed sampling strategy can well reconstruct Group-1 Element 4, but the image reconstructed by the polynomial sampling strategy appears a little bit noisy. When the sampling ratio is 10%, the circular sampling strategy can reconstruct Group-1 Element 5. The polynomial sampling strategy and the proposed sampling strategy can well reconstruct Group 0 Element 2.

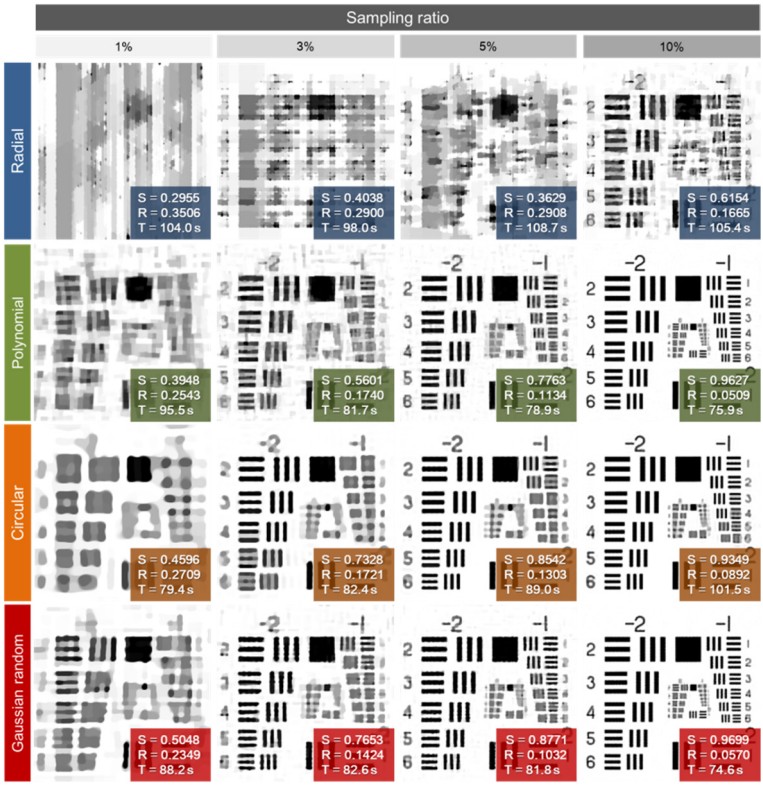

**Figure 4.** Comparison of the reconstruction results of USAF-1951 resolution test chart for different sampling strategies and sampling ratios. The SSIM, RMSE, and image reconstruction time (denoted by S, R, and T, respectively) are given in the inset of each reconstruction.

We also quantitatively evaluate the reconstruction quality by using structural similarity index (SSIM) [37] and root-mean-square error (RMSE). The SSIM, RMSE, and image reconstruction time (denoted by S, R, and T, respectively) are given in the inset of each reconstruction. The quantitative measures also demonstrate the proposed sampling strategy has better performance than the other sampling strategies in comparison.

In the second simulation, we use a natural image—"Cameraman"—for testing. The size of the test image is also 256 × 256 pixels. As the results show in Figure 5, the images reconstructed by the radial sampling strategy turn out to be the worst, especially when the sampling ratio is lower than 10%. The polynomial, the circular, and the proposed sampling strategies are capable of reconstructing recognizable contents when the sampling ratio is 3%. The image reconstructed by the proposed method appears clearer, which is evident by the details (the cameraman's face, the camera, and the buildings) that are reconstructed. Both the SSIMs and RMSEs demonstrate the advantage of the proposed method.

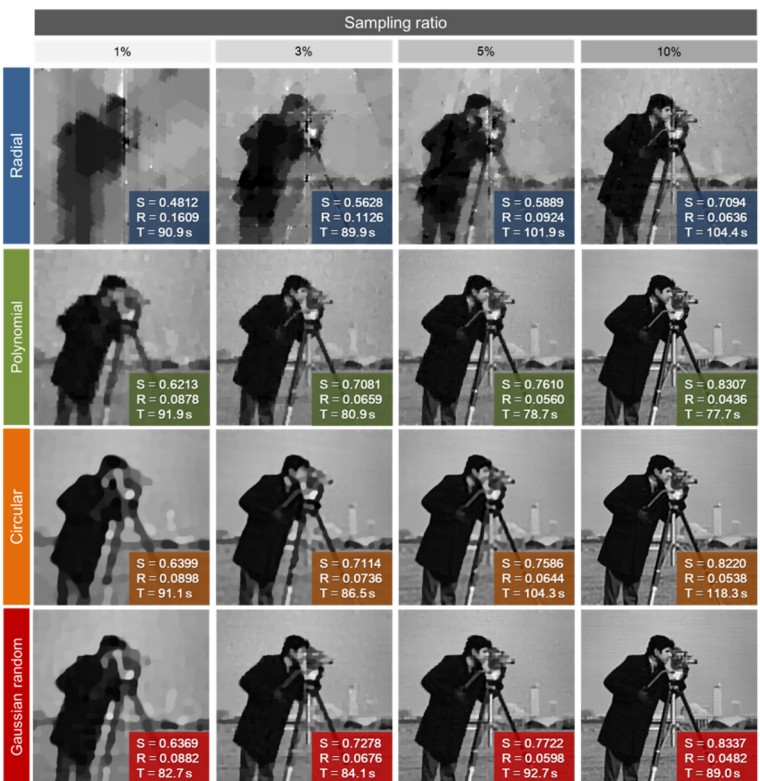

**Figure 5.** Comparison of the reconstruction results of "Cameraman" for different sampling strategies and sampling ratios. The SSIM, RMSE, and image reconstruction time (denoted by S, R, and T, respectively) are given in the inset of each reconstruction.

## 4. Experiment

The proposed method is also demonstrated with real experiments. The schematic diagram of the experimental set-up is shown in Figure 1a. The set-up consists of a 12-watt white-light LED, a DMD (ViALUX V-7001), an imaging lens, a target object, a collecting lens, and a PDA (Thorlabs PDA101A). Note that we binarize the Fourier basis patterns with the upsample-and-dither strategy [29], so as to take advantage of the high-speed binary pattern generation offered by the DMD. The patterns are initially with $256 \times 256$ pixels. The patterns are upsampled with a ratio of 2 through the bicubic interpolation and then binarized using the Floyd–Steinberg algorithm [33]. We use two different scenes for the experiment. The one scene is a USAF-1951 resolution target printed on a piece of A4 paper. The other scene consists of a pair of china dolls as foreground and the printed resolution target pattern as background.

Similarly, we compare the proposed Gaussian random sampling strategy with the radial, the circular, and the polynomial sampling strategies in the experiments. As the reconstructed images show in Figure 6, the experimental results coincide with the simulation results, demonstrating that the proposed sampling strategy allows for better imaging quality especially when the sampling ratio is low. Please note that the reconstructed images presented in Figure 6 are acquired at the DMD rate of 50 Hz.

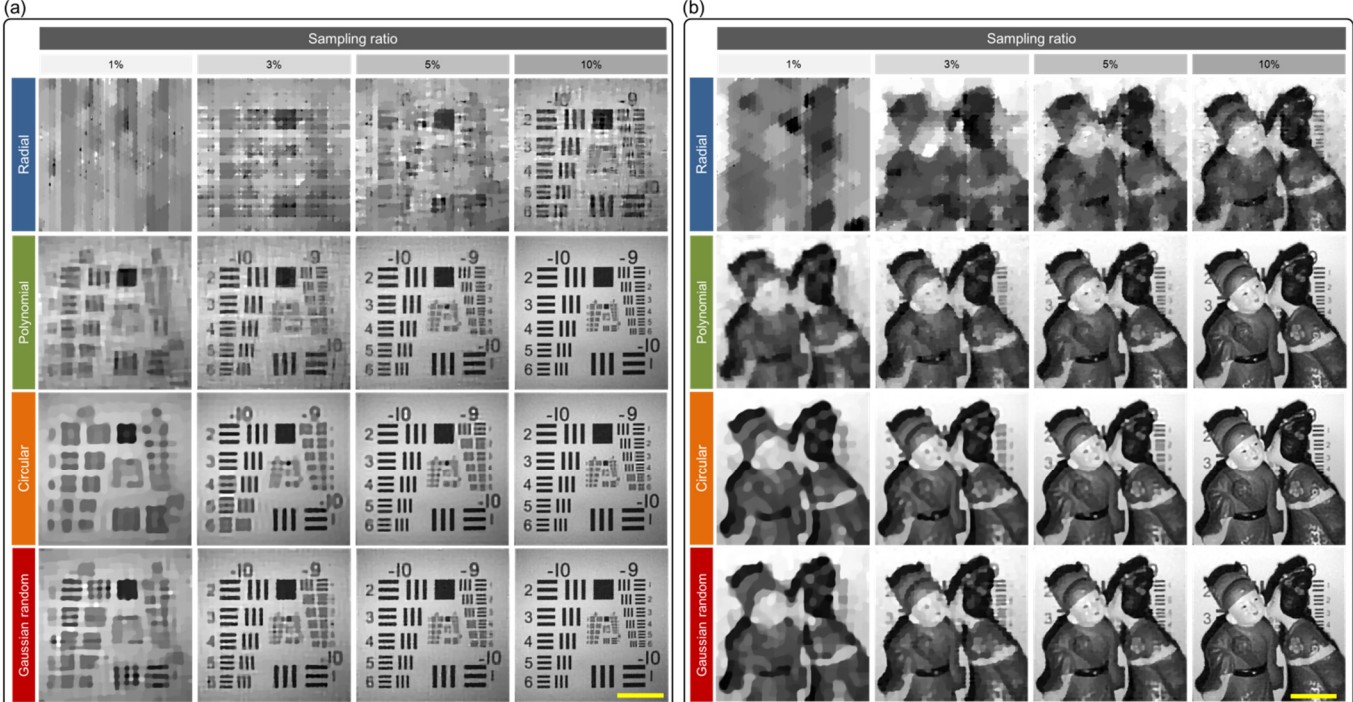

**Figure 6.** Experiment results for (**a**) USAF-1951 resolution test chart pattern and (**b**) a pair of china dolls. DMD refreshing rate is 50 Hz. Scale bars = 5 cm.

In order to demonstrate the proposed fast single-pixel imaging, we test the proposed method with different DMD refreshing rates. In this test, the sampling ratio is set to 10%, and therefore, the number of single-pixel measurements is 9831. As the results show in Figure 7, the reconstructions for 50 Hz and 200 Hz are clear and without noticeable noise. In other words, the proposed method is able to capture a high-quality image of $256 \times 256$ pixels within 50 s. As the DMD refreshing rate increases, the noise becomes obvious and the signal-noise ratio (SNR) decreases. The image for 2000 Hz is slightly noisy, but the data acquisition time can be reduced down to 5 s. When the DMD refreshing rate is 20,000 Hz, the image appears noisy, but the objects in the image are still recognizable.

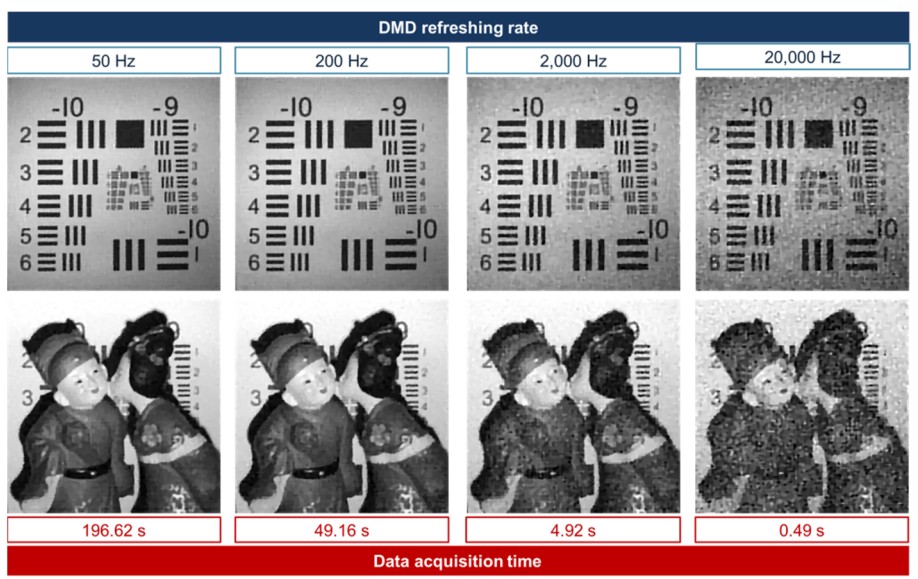

**Figure 7.** Experiment results reconstructed by the proposed method for different DMD refreshing rates. Sampling ratio = 10%.

## 5. Discussion

The total imaging time in single-pixel imaging includes data acquisition time and the image reconstruction time. This work aims at improving the sampling efficiency in FSI to reduce the data acquisition time. For the purpose of imaging a dynamic scene, a short data acquisition time is desirable, because the data acquisition time in single-pixel imaging is like the exposure time in conventional photography. Severe blur might be caused by the motion of objects if the data acquisition time is long. We note that the proposed sampling strategy requires CS for image reconstruction and CS algorithms are commonly computationally exhausted. In our future work, we consider using deep learning [38–45] to reconstruct the final image from the undersampled Fourier spectrum so as to reduce the image reconstruction time.

In this paper, we demonstrate that the proposed Gaussian random sampling strategy can effectively improve the sampling efficiency of FSI. We consider that the proposed sampling strategy is applicable to other basis-scan single-pixel methods. The core of the proposed sampling strategy is to conduct density-varying sampling in an orthogonal transform domain so as to improve sampling efficiency for fast single-pixel imaging. The proposed sampling strategy utilizes the fact that the energy of any natural image concentrates at the low-frequency band of a certain transform domain. It is such ununiformity of energy distribution that enables density varying sampling. Natural images have sparse representation in DCT, Hadamard transform, and wavelet transform domains. We therefore consider that the reported sampling strategy can be applied in DCT single-pixel imaging, Hadamard single-pixel imaging, and other basis scan single-pixel imaging methods.

In comparison with the polynomial sampling strategy, the proposed method can reproduce better images when the sampling ratio is low. In addition, the proposed method requires no user-defined parameters, which adds flexibility to the method in practical use.

## 6. Conclusions

We propose the Gaussian random sampling strategy to achieve efficient FSI. The key to the proposed sampling strategy is to conduct density-varying sampling in the Fourier domain so as to improve sampling efficiency for fast single-pixel imaging. As is demonstrated by the simulations and the experiments, the proposed method is able to reproduce a sharp and clear image of $256 \times 256$ pixels with a sampling ratio of 10%. This work benefits fast single-pixel imaging and provides a new approach for efficient spatial information acquisition.

**Author Contributions:** Conceptualization, Z.Z. and J.Z.; validation Z.Q., X.G., T.L. and P.Q.; funding acquisition, Z.Z. and J.Z.; writing, Z.Z. and Z.Q.; supervision, J.Z. All authors have read and agreed to the published version of the manuscript.

**Funding:** This research was funded by National Natural Science Foundation of China (NSFC), (61905098 and 61875074) and Guangzhou Basic and Applied Basic Research Foundation (202002030319 and 202102080589).

**Data Availability Statement:** The data presented in this study are available on request from the corresponding author.

**Conflicts of Interest:** The authors declare no conflict of interest.

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
