# Peer review of "Efficient Fourier Single-Pixel Imaging with Gaussian Random Sampling"

_photonics, doi:10.3390/photonics8080319_

Round 1

Reviewer 1 Report

Qiu et al. reported a sampling strategy for efficient Fourier single-pixel imaging. The authors claimed that the reported sampling strategy can well cooperate with compressive sensing algorithms to reconstruct a clear and sharp image with a reduced number of single-pixel measurements. The key of the reported sampling strategy is to utilize the property of energy concentration at the low-frequency band of natural images and conduct a variant density sampling in the Fourier domain. Specifically, the lower frequency band is sampled with a higher density while the high-frequency band with a lower density. The density is subject to a Gaussian function. The unsampled spectral information is recovered through optimization, that is, compressive sensing. Both simulations and experiments demonstrate the effectiveness of the reported method. I have several questions about the method reported.

  1. The authors have validated the reported Gaussian random sampling strategy in Fourier single-pixel imaging. Can the authors discuss the applicability of the reported sampling strategy in other single-pixel imaging methods?
  2. The reported sampling strategy relies on a pre-generated sampling path in the Fourier domain. As Fig. 1 shows, the sampling path is for images of 256-by-256 pixels. What if the image to be reconstructed is of other sizes, such as, 256-by-192, 512-by-512? Is the reported method still applicable in this case?

Author Response

Please see the attached PDF document.

Reviewer 2 Report

To reduce the data acquisition time of Fourier single-pixel imaging, the authors proposed a new sampling strategy for FSI with Gaussian random sampling. However, I have some questions.

  1. Have you ever read the paper “Sparse Fourier single-pixel imaging. Optics Express (2019)”, where an extremely similar variable density random sampling matrix is employed to achieve random sampling with Fourier single-pixel imaging technology, followed by using compressive sensing algorithms to recover the high-quality information of the object? So, what’s the difference between the method proposed earlier with yours? Does your method work better than his, or does it have any advantages? Please give the relevant contrast simulations and experiments.
  2. By using an advanced CS algorithm, the image reconstruction time is increased. Does that mean the whole imaging time is invariant or even increased? However, if you use deep learning, even though the training process spends much time, the whole imaging time is much short with less measurements and IFT2, so what is the significance of your method for studying fast Fourier single pixel imaging with high imaging quality? Please elaborate it.

Author Response

Please see the attached PDF document.

Reviewer 3 Report

The autors report a Fourier single pixel imaging method based on random sampling in the Fourier plane. The general proposed technique may be  possibly of interest in particular because due to the combined contributions of compressive sensing algorithms and short acquisition time.  However the major remarks and questions after reading are the following : 

- The authors must clearly develop and explain the part of the paper relevant to step phase shifting and compressive sensing algorithms which are used for image reconstruction. This  leads to optimize the choice of the high importance coefficients in the spectrum. Also which are the main parameters to consider for the selection of the radial and circular sampling (185)  also line (191) comparaison on the used CS algorithms  and their influence on the quality of image reconstruction. 

- Which is the  significance of the similarity index in line 223. In fig 7 -8 quantify the spatial resolution and the noise for the optimum sampling strategy and sampling ratio. 

- The target image is 256x256 pixels. Can the method be extended to much higher number of pixels while maintaining good spatial resolution and processing time ?  

To conclude the paper has interest as a fairly novel technique for single pixel imaging, but a revised  form of the manuscript is required, it must clearly develop and analyse the principles of the mask sampling and compressive sensing.   

Author Response

Please see the attached PDF document.

Reviewer 4 Report

The manuscript of Qiu et al. presents the interesting approach of the use of The Gaussian function as a weight function for acquisition bands in Fourier single-pixel imaging. However, before the publication, some corrections to the manuscript should be made.

  • In general, it is difficult to determine the authors' contributions to the development of the FSI from the current version of the manuscript. Apart from proposing the use of the Gauss function as an acquisition band weighting function, the proposed approaches are based on the assumptions of other authors regarding Fourier component ranking [18],or the use of CS algorithm, L1-Magic algorithms [42]. The authors should describe in more detail their contribution to the development of this computational imaging technique.
  • In Fig.1a, the Authors are showing the optical system setup for FSI, where they are using the structured illumination approach to illuminate the object by the sinusoidal intensity patterns. How this condition was incorporated into the simulation based on the image database? In the experimental results, the authors did not provide information about the period, the spatial frequency of the illumination patterns projected on the examined objects.
  • Authors should provide the comparison of the time of image recovery by the proposed algorithm and other alternatives?
  • The Authors described that they used different masks for sampling the Fourier space and determining the acquisition band. One of them was the Gaussian mask. Why it does not have a uniform distribution in all directions? Are the maxima similar for sinc function can be observable along with fx, fy axis? It is the Gaussian function or its Fourier spectrum?
  • Since the Authors examined the image recovery/reconstruction, they should perform a more extended comparison of the obtained results and the original images. As it is most common, they should calculate the recovery/reconstruction errors describing the differences between two images for example in terms of root-mean-square error (RMSE).
  • What quantities ( in colour rectangles) are added to each reconstructed image on Fig. 4 ( t is resolution? Please specify the units) or it is the structural similarity index? The authors indicated that they "quantitatively evaluate the results by using structural similarity index(SSIM)" (Line 223), but they did not define this parameter and did not present the results, or they were not directly indicated in figures' captions.
  • What objective criterion of the appropriate reconstruction of the target object image ( USAF-1951) was used? How objectively or quantitatively the authors decided which test was appropriately reconstructed? Did they analyse the contrast between test lines?
  • Can the Authors include the table with the comparison of the resolution possible to be obtained by their algorithm, SSIM value, time for each approach and sampling ratio? This comment is related to both simulation and experimental results.
  • Since the proposed approach is describing the computational imaging algorithms which performance is depending on the used PC, the detailed technical specification of the PC should be provided.
  • Moreover, the authors should provide the information about used software?

Author Response

Please see the attached PDF document.

Round 2

Reviewer 3 Report

The authors have fairly well developed the questions relevant to the novel sampling mask method and data acquisition time. The scientific content is  improved and it includes more datas to justify the interest of the methods. 

The paper can now be accepted for publication in the journal